## Methods and techniques

**Subject Area:**
biochemistry/biophysics/cellular biology/
molecular biology

mass spectrometry, RNA cap, RNA methylation, RNA processing, 7-methylguanosine, ribose O-2 methylation

**Authors for correspondence:**
Michael A. J. Ferguson
e-mail: m.a.j.ferguson@dundee.ac.uk
Victoria H. Cowling
e-mail: v.h.cowling@dundee.ac.uk

# CAP-MAP: cap analysis protocol with minimal analyte processing, a rapid and sensitive approach to analysing mRNA cap structures

Alison Galloway[1], Abdelmadjid Atrih[2], Renata Grzela[4], Edward Darzynkiewicz[4,5], Michael A. J. Ferguson[3] and Victoria H. Cowling[1]

[1]Centre for Gene Regulation and Expression, School of Life Sciences, [2]FingerPrints Proteomics Facility, School of Life Sciences, and [3]Wellcome Centre for Anti-Infectives Research, School of Life Sciences, University of Dundee, Dundee DD1 5EH, UK
[4]Centre of New Technologies, University of Warsaw, and [5]Division of Biophysics, Institute of Experimental Physics, Faculty of Physics, University of Warsaw, 02-097 Warsaw, Poland

  AG, 0000-0001-8780-0382; VHC, 0000-0001-7638-4870

Eukaryotic messenger RNA (mRNA) is modified by the addition of an inverted guanosine cap to the 5′ triphosphate. The cap guanosine and initial transcribed nucleotides are further methylated by a series of cap methyltransferases to generate the mature cap structures which protect RNA from degradation and recruit proteins involved in RNA processing and translation. Research demonstrating that the cap methyltransferases are regulated has generated interest in determining the methylation status of the mRNA cap structures present in cells. Here, we present CAP-MAP: cap analysis protocol with minimal analyte processing, a rapid and sensitive method for detecting cap structures present in mRNA isolated from tissues or cultured cells.

## 1. Introduction

Eukaryotic RNA polymerase II (RNAPII)-transcribed RNAs are modified by the addition of 7-methylguanosine to the 5′ triphosphate found on the first transcribed nucleotide, forming the cap structure denoted $^{m7}$GpppN (N is any nucleotide) (figure 1) [1,2]. For many short RNAPII transcripts involved in guiding RNA processing and modification, the cap is a precursor for further modification [3]. For pre-messenger RNA (mRNA), the cap guides transcript processing and selection for translation via interactions with cap-binding proteins, while protecting the transcript from 5′–3′ exonucleases [4]. Metazoan mRNA caps additionally contain ribose methylated on the O-2 position of the first and second transcribed nucleotides (denoted $N_m$), which creates a further 'self-mRNA' mark, enabling innate immunity proteins to differentiate it from unmethylated, foreign RNA [5,6]. Removal of the cap by decapping enzymes usually directs mRNA to be degraded. Decapping is also regulated by RNA binding proteins which promote or antagonize the recruitment of decapping complexes to the mRNA [7]. Thus, the cap is essential for the proper processing, function and lifespan of an mRNA.

 Mammalian mRNA processing initiates co-transcriptionally with the addition of the inverted guanosine cap, catalysed by Capping Enzyme/RNA guanylyltransferase and 5′ phosphatase (CE/RNGTT) [2,8]. The terminal cap guanosine is methylated by RNA cap methyltransferase (RNMT) on the N-7 position and the first two transcribed nucleotides are modified by cap-specific methyltransferases as follows: the O-2 position of the ribose of the first and second transcribed nucleotides are methylated by Cap Methyltransferase 1 (CMTR1) and CMTR2,

**Figure 1.** mRNA cap structure. A common cap structure is depicted, including cap guanosine, the first transcribed nucleotide and the second transcribed nucleotide. The sites of action of the capping enzymes RNGTT, RNMT, CMTR1, CMTR2 and CAPAM are indicated.

respectively, and if the first transcribed nucleotide is adenosine it is methylated on the N-6 position by cap-specific adenosine methyltransferase (CAPAM, also known as PCIF1) [9–12] (figure 1).

The cap structure was first discovered in viral mRNA and determined to mimic eukaryotic cellular mRNA in order to hijack the translation machinery and evade detection by the host innate immunity proteins [1]. From the 1970s, $^{m7}GpppN_{m}$-pN and $^{m7}GpppN_{m}pN_{m}$ caps, as well as the $^{m6}A_{m}$ modification, were detected in mammalian cells by chromatographically separating RNAse-digested radiolabelled mRNA and by mass spectrometry (MS) [13–19]. More recently, it has become apparent that the cap methyltransferases are regulated during important cellular processes including the cell cycle, oncogenic transformation, anti-viral responses and embryonic stem cell differentiation [20–26]. This has rejuvenated interest in determining the relative abundance of each cap structure and additional cap analysis techniques have been developed [27]. First nucleotide ribose O-2 methylation and adenosine N-6 methylation have been detected by radiolabelling the terminal nucleotides and resolving them by two-dimensional thin-layer chromatography (TLC) [28]. Since this method does not require *in vivo* labelling, mRNA from any source can be analysed and the method can be adapted to investigate methylation of specific mRNAs; however, it is technically challenging and requires the use of radio-isotopes. Affinity reagents can be used to enrich methylated nucleotides and specific cap structures in a semi-quantitative manner. Capped mRNA can be enriched using antibodies that recognize $^{m2,2,7}G$ and $^{m7}G$, although such antibodies exhibit gene specificity which may be due to the cap structures present or RNA secondary structure [24,29,30]. Recent studies have detected internal $^{m7}G$ in mRNA, which compromises the use of anti-7-methylguanosine antibodies for cap detection [31,32]. Similarly, internal and cap-adjacent $^{m6}A$ have been mapped using an anti-$^{m6}A$ antibody, but since internal $^{m6}A$ is abundant this method requires knowledge of the transcription start site to determine $^{m6}A$-containing cap structures. [33,34]. Recombinant eIF4E can also be used as an affinity reagent for capped mRNA although this does not specify particular cap structures [4]. Recently, an antibody-free $^{m6}A$ mapping method has been developed [35]. A cytidine deaminase fused

to the $^{m6}A$-binding domain of the YTH domain was expressed in cells, resulting in C-to-U deamination at sites adjacent to $^{m6}A$ residues which can be detected by RNA-seq. With the advent of nanopore sequencing, methylated nucleotides can be detected on specific sequences and this is likely to include the first transcribed nucleotides of the cap in the near future [36].

MS has been employed to detect various nucleotide modifications, including the cap [1,37,38]. Since the first transcribed nucleotide is expected to be $N_{m}$ (ribose O-2 methylated), the presence of $^{m6}A_{m}$ is associated with cap structures. A more accurate method for cap analysis involves detecting the cap dinucleotide $^{m7}GpppN_{m}$, or short 5′ mRNAs including this structure. Recently, this has been used successfully to detect cap structures derived from short RNAs, precursor tRNAs, as well as to monitor the activities of CAPAM and the $^{m6}A$ demethylase fat mass and obesity-associated protein (FTO) on the first transcribed nucleotide of *in vitro* transcribed RNA [3,25,39–41]. More recently, cellular mRNA cap structures isolated with an anti-$^{m7}G$ antibody have been analysed by MS and $^{m7}GpppG_{m}G_{p}$, $^{m7}GpppA_{m}G_{p}$ and $^{m7}Gppp^{m6}A_{m}$-$G_{p}$ caps were readily detected [9]. Caps lacking first transcribed nucleotide methylation were not detected, indicating their rarity.

To gain a better understanding of *in vivo* cap regulation, we developed an approach to determine the relative proportions of the different mRNA cap structures using a rapid and unbiased approach. Here, we present the CAP-MAP (cap analysis protocol with minimal analyte processing) method for detecting and quantifying mRNA cap structures by liquid chromatography–mass spectrometry (LC–MS).

## 2. Results

We developed CAP-MAP to detect and quantify the mRNA cap structures present in cells and tissues. Specifically, the method assesses the permutations of N-7 methylation of the terminal cap guanosine, O-2 methylation of the first nucleotide ribose and N-6 methylation of the first nucleotide adenosine in mRNA caps from biological samples (figure 1). Briefly, the method involves mRNA enrichment from total cellular RNA using oligo dT affinity beads. The

royalsocietypublishing.org/journal/rsob    Open Biol. **10**: 190306

**Figure 2.** Overview of RNA preparation for CAP-MAP analysis. Cellular RNA is purified on oligo dT-conjugated beads and digested with P1 nuclease to release cap dinucleotides and nucleotide monophosphates. The synthetic cap standard, ARCA, is added to the digested nucleotides. The sample is run on a PGC column coupled to a triple quadrupole mass spectrometer operating in negative ion mode and programmed to detect cap dinucleotides in the MRM mode.

mRNA is digested with nuclease P1, a non-specific ssRNA/DNA nuclease, to release nucleotide 5′ monophosphates and cap dinucleotides, $^{(m7)}GpppN_{(m)}$. The latter is resolved by liquid chromatography on a porous graphitic carbon (PGC) column and identified by negative ion electrospray MS using multiple reaction monitoring (MRM) (figure 2). Negative ion mode was selected because of the propensity of nucleotides to form negative ions and for its better signal-to-noise ratio relative to positive ion mode. MRM is a procedure whereby, at any given moment, the first quadrupole/mass filter is programmed to allow through ions of only a single $m/z$ value (X) into the second quadrupole/collision cell, where collision-induced dissociation (CID) occurs, and the third quadrupole/mass filter is programmed to allow through ions of typically only two $m/z$ values (Y and Z) to the detector. In this way, ion currents (peaks) are only recorded when an analyte that produces a precursor ion (in this case an $[M-H]^-$ ion) of $m/z$ X generates product ions of $m/z$ Y and Z. This, together with the LC retention time, provides confidence in the analyte identification as well as great sensitivity since the mass spectrometer is not spending time scanning irrelevant mass ranges.

## 2.1. LC–MS analysis of cap dinucleotides

Eleven synthetic cap dinucleotides, four obtained commercially and seven custom synthesized (table 1), were used to established retention times relative to an internal standard on PGC liquid chromatography and to optimize MRM conditions (mass transitions and collision energies) for their detection and quantification (table 2). These cap dinucleotides have adenosine or guanosine as the first transcribed nucleotide, thus our method will only detect these variants and not those with cytidine or uridine, which are also present in cells. The internal standard was the anti-reverse cap analogue (ARCA), which is not present in cells. ARCA has the structure $^{m7}G_{O-3\ m}pppG$, which is physically similar to but structurally distinct from endogenous cellular caps. During method development, we selected HyperCarb PGC as the preferred LC column type over SeQuant ZICpHILIC because the latter could not resolve the isobaric cap structures $^{m7}Gpppm^{6}A$ and $^{m7}GpppA_m$. Using PGC, we found that excellent peak shape and low carry-over between runs (less than 0.1%) could be obtained by using a pH 9.15 aqueous component and maintaining the column at 45°C. Furthermore, the high organic solvent (acetonitrile) content required for cap elution

**Table 1.** Synthetic cap dinucleotide standards.

| cap | source |
|---|---|
| GpppA | NEB |
| $^{m7}$GpppA | NEB |
| GpppG | NEB |
| $^{m7}$GpppG | NEB |
| Gppp$^{m6}$A$_m$ | synthesized |
| $^{m7}$Gppp$^{m6}$A$_m$ | synthesized |
| GpppA$_m$ | synthesized |
| GpppG$_m$ | synthesized |
| $^{m7}$GpppG$_m$ | synthesized |
| $^{m7}$GpppA$_m$ | synthesized |
| $^{m7}$Gppp$^{m6}$A | synthesized |
| ARCA ($^{m7}$G$_{0-3}$ $_m$pppG) | NEB |

from PGC was advantageous in the electrospray ionization process. Using the conditions described in the Materials and methods, all 11 cap dinucleotides could be resolved by retention time and/or MRM transitions (figure 3). To reduce PGC column deterioration over time, leading to peak broadening and increasing retention times [42,43], we adopted some previous recommendations [43,44]. In particular, we regularly regenerated the columns by equilibrating in 95% methanol for between 1 h and 16 h.

To determine the linear range of detection for the cap dinucleotides during LC–MS, a dilution series of 10 of the cap dinucleotides was analysed (figure 4a,b). All cap dinucleotides were detected in a linear range down to approximately 4 fmol, with detection of some caps being nonlinear at lower concentrations. Linear regression was used to calculate the slope for the peak area/fmol of each cap, which was later used to convert peak area to fmol quantities for each cap (table 3).

$^{m7}$GpppG and GpppG$_m$ caps are a challenge to distinguish by LC–MS because they are isobaric, elute very closely on PGC and produce similar CID spectra. Fortunately, $^{m7}$GpppG generates a unique product ion at $m/z$ 635.9; thus we can detect and quantify $^{m7}$GpppG with the unique precursor → product ion transition of $m/z$ 800.9 → 635.9. However, both $^{m7}$GpppG and GpppG$_m$ produce precursor → product ion transitions of $m/z$ 800.9 → 423.9 and 438.0. Thus, to quantify GpppG$_m$ we have to correct for any $m/z$ 800.9 → 423.9 and 438.0 transition ion currents due to $^{m7}$GpppG . This is possible because the ratio of $m/z$ 635.9 : 423.9 : 438 product ion intensities for $^{m7}$GpppG are constant. Consequently, the $m/z$ 423.9 and 438 product ion contributions due to $^{m7}$GpppG can be back-calculated from the $^{m7}$GpppG-unique $m/z$ 635.9 ion current. Examples of the calculations of the concentrations of $^{m7}$GpppG and GpppG$_m$ in liver samples are shown in figure 4e.

## 2.2. LC–MS analysis of mRNA cap dinucleotides from liver

Having established an LC–MS protocol with synthetic cap dinucleotides we proceeded to analyse cellular mRNA cap structures. Digestion of cellular mRNA with nuclease P1 results in a mixture of cap dinucleotides, nucleotide monophosphates

and the nuclease protein. Various strategies including weak anion exchange and precipitation protocols were trialled to enrich and concentrate cap dinucleotides; however, these attempts resulted in a substantial loss of sample. We were also concerned that the enrichment methods may selectively alter the composition of cap dinuclelotides in a sample. Therefore, we analysed cellular cap dinucleotides by LC–MS with minimal processing by directly injecting the whole nuclease P1 digest. Since there was no purification of the cap nucleotides after digestion, there was a risk that the nuclease P1 might accumulate on the column, increasing the back pressure. However, the back pressure of the column was monitored over 100 runs and was stable at around 40 bar.

When we analysed the mRNA cap structures present in mouse liver, we were able to detect nine of the 11 cap dinucleotides assessed (figure 5a). Importantly, we found that the absolute ion currents recorded for the ARCA internal standard were the same whether or not it was injected alone or with cellular mRNA samples, suggesting no detectable interference (ion suppression) from the cellular mRNA nuclease P1 digests. Nucleotide monophosphates make up the majority of the nuclease P1 digests; adenosine monophosphate (AMP) eluted before cap dinucleotides (electronic supplementary material, figure S1).

The most common cap structures in liver mRNA were $^{m7}$GpppG$_m$ (19 fmol/µg), $^{m7}$GpppA$_m$ (4 fmol/µg) and $^{m7}$Gppp$^{m6}$A$_m$ (20 fmol/µg) (figure 5a); these are the major cap variants previously elucidated by MS and TLC of radio-labelled nucleotides [1,28]. In line with recent observations that $^{m7}$Gppp$^{m6}$A$_m$ is an abundant cap structure, $^{m7}$Gppp$^{m6}$A$_m$ constituted 75% of the caps when A was the first transcribed nucleotide [9–12]. A number of incompletely methylated cap structures were detected including GpppA, GpppA$_m$, Gppp$^{m6}$A$_m$ and GpppG$_m$, which lack cap guanosine N-7 methylation; and $^{m7}$GpppA and $^{m7}$GpppG, which lack ribose O-2 methylation of the first transcribed nucleotide.

To determine the minimum input of mRNA required to detect the different cap structures, we prepared a dilution series of liver mRNA. Ten cap dinucleotides, including in this experiment GpppG, were detected in 30 µg and in 15 µg mRNA, and the three most abundant caps were reproducibly detected in 250 ng mRNA (figure 5b,c). The concentration (fmol µg$^{-1}$) of each cap dinucleotide was consistent across a range of inputs, when above the threshold of detection for that cap. The consistent detection of mRNA caps and ARCA internal standard across a range of input mRNA is further evidence of insignificant ion suppression from the cellular mRNA nuclease P1 digests.

In summary, we established CAP-MAP as a rapid, quantitative and relatively direct method for detecting mRNA cap dinucleotides.

## 2.3. CAP-MAP comparison of mRNA caps from mammalian cells and tissues

To compare the relative cap abundancies in tissues, mRNA samples were prepared from liver, heart, brain and activated CD8 T cells. In each sample, the most common cap dinucleotides were $^{m7}$GpppG$_m$ and $^{m7}$Gppp$^{m6}$A$_m$, which were present at greater than 10 fmol µg$^{-1}$ across all sources (figure 6a and table 4). $^{m7}$GpppA$_m$ was less common, but always present over 0.5 fmol µg$^{-1}$ mRNA. A variety of less

royalsocietypublishing.org/journal/rsob    Open Biol. **10**: 190306

**Table 2.** Mass spectrometer settings.

| compound | precursor (m/z) | product (m/z) | collision energy (V) | RF lens (V) | retention time relative to ARCA |
|---|---|---|---|---|---|
| GpppA | 771.1 | 408.0 | 36 | 150 | 0.95 |
| GpppA | 771.1 | 424.0 | 36 | 150 | 0.95 |
| GpppA$_m$ | 785.0 | 422.0 | 35 | 123 | 0.95 |
| GpppA$_m$ | 785.0 | 424.0 | 34 | 123 | 0.95 |
| $^{m7}$GpppA | 785.0 | 407.9 | 37 | 103 | 0.98 |
| $^{m7}$GpppA | 785.0 | 620.0 | 26 | 103 | 0.98 |
| GpppG | 787.1 | 424.0 | 38 | 160 | 0.94 |
| GpppG | 787.1 | 441.9 | 37 | 160 | 0.94 |
| Gppp$^{m6}$A$_m$ | 799.0 | 423.6 | 34 | 126 | 1.05 |
| Gppp$^{m6}$A$_m$ | 799.0 | 436.0 | 36 | 126 | 1.05 |
| $^{m7}$GpppA$_m$/$^{m7}$Gppp$^{m6}$A | 799.0 | 422.0 | 37 | 126 | 0.98/1.08 |
| $^{m7}$GpppA$_m$/$^{m7}$Gppp$^{m6}$A | 799.0 | 633.8 | 26 | 126 | 0.98/1.08 |
| $^{m7}$GpppG | 800.9 | 635.9 | 28 | 120 | 0.97 |
| GpppG$_m$ | 801.0 | 423.9 | 36 | 119 | 0.94 |
| GpppG$_m$ | 801.0 | 438.0 | 38 | 119 | 0.94 |
| $^{m7}$Gppp$^{m6}$A$_m$ | 813.0 | 436.0 | 39 | 108 | 1.08 |
| $^{m7}$Gppp$^{m6}$A$_m$ | 813.0 | 438.0 | 35 | 108 | 1.08 |
| $^{m7}$GpppG$_m$ | 815.0 | 438.0 | 39 | 112 | 0.96 |
| $^{m7}$GpppG$_m$ | 815.0 | 517.9 | 37 | 112 | 0.96 |
| ARCA | 815.2 | 423.9 | 32 | 130 | 1.00 |
| ARCA | 815.2 | 451.9 | 32 | 130 | 1.00 |

common cap dinucelotides were detected in the tissues and T cells; GpppA$_m$, $^{m7}$GpppG and GpppG$_m$ were detected above 0.1 fmol µg$^{-1}$ in all tissues; GpppA was present at approximately 0.5 fmol µg$^{-1}$ in the heart and liver, but inconsistently detected in other tissues; $^{m7}$GpppA was present in low amounts in all samples except for one of the brain samples, in which it was not detected; and Gppp$^{m6}$A$_m$ was very low in abundance and only consistently detected in liver (figure 6*b*). Overall the minor cap variants, which lack either cap guanosine N-7 methylation or first nucleotide ribose O-2 methylation, constituted around 2–5% of the cellular caps.

Among the more common cap dinucleotides there were tissue-specific differences (figure 6*a,b*). In most samples, the abundances of caps with the first transcribed nucleotide A and G were similar; however, in the heart, there were more than twice as many caps with A as the first transcribed nucleotide as there were with G as the first transcribed nucleotide. This likely reflects differences in the heart transcriptome, compared with other tissues analysed. The other major difference in cap structure abundance between tissues was the proportion of first nucleotide adenosine caps with the $^{m6}$A modification. All tissues had a greater concentration of $^{m7}$Gppp$^{m6}$A$_m$ than $^{m7}$GpppA$_m$, but the ratio of these two cap structures varied considerably between tissues. The liver had the lowest $^{m7}$Gppp$^{m6}$A$_m$ : $^{m7}$GpppA$_m$ ratio of 3.5. Brain, at the other extreme, had an $^{m7}$Gppp$^{m6}$A$_m$ : $^{m7}$GpppA$_m$ ratio of 14.6 (figure 6*c*). These results indicated tissue-specific regulation of the $^{m6}$A cap methyltransferase, CAPAM, and/or the demethylase FTO. To demonstrate that our method can

detect changes in the mRNA cap structure following perturbations in cap methyltransferase activity we reduced CAPAM expression by transfection of siRNA in HeLa cells. CAPAM knockdown was confirmed by western blotting (figure 7*a*). As expected, CAPAM knockdown results in an increase in $^{m7}$GpppA$_m$ and a decrease in $^{m7}$Gppp$^{m6}$A$_m$ (figure 7*b* and table 4). In the control siRNA transfected HeLa cells, there is about ninefold more $^{m7}$Gppp$^{m6}$A$_m$ than $^{m7}$GpppA$_m$, whereas this ratio drops to about 1.2-fold when CAPAM expression is repressed (figure 7*c*). Notably, the amount of $^{m7}$GpppG$_m$ which is not a CAPAM substrate or product is very similar between the control and CAPAM siRNA-treated cells. These findings demonstrate that short-term regulation of a cap methyltransferase causes sufficient changes in mRNA structures to be detectable by CAP-MAP.

## 3. Discussion

Here, we report the development of CAP-MAP, a rapid, sensitive and relatively direct method of mRNA cap analysis. The benefit of CAP-MAP is that, following oligo dT-based enrichment of mRNA and digestion, cap structures are ready for LC–MS analysis. This simple preparation is possible owing to the selectivity of the PGC column and unique triphosphate-linked dinucleotide structure of the mRNA caps. The small number of preparative steps means that the relative proportion of cellular cap structures is likely to be faithfully maintained during analysis.

royalsocietypublishing.org/journal/rsob    Open Biol. **10**: 190306

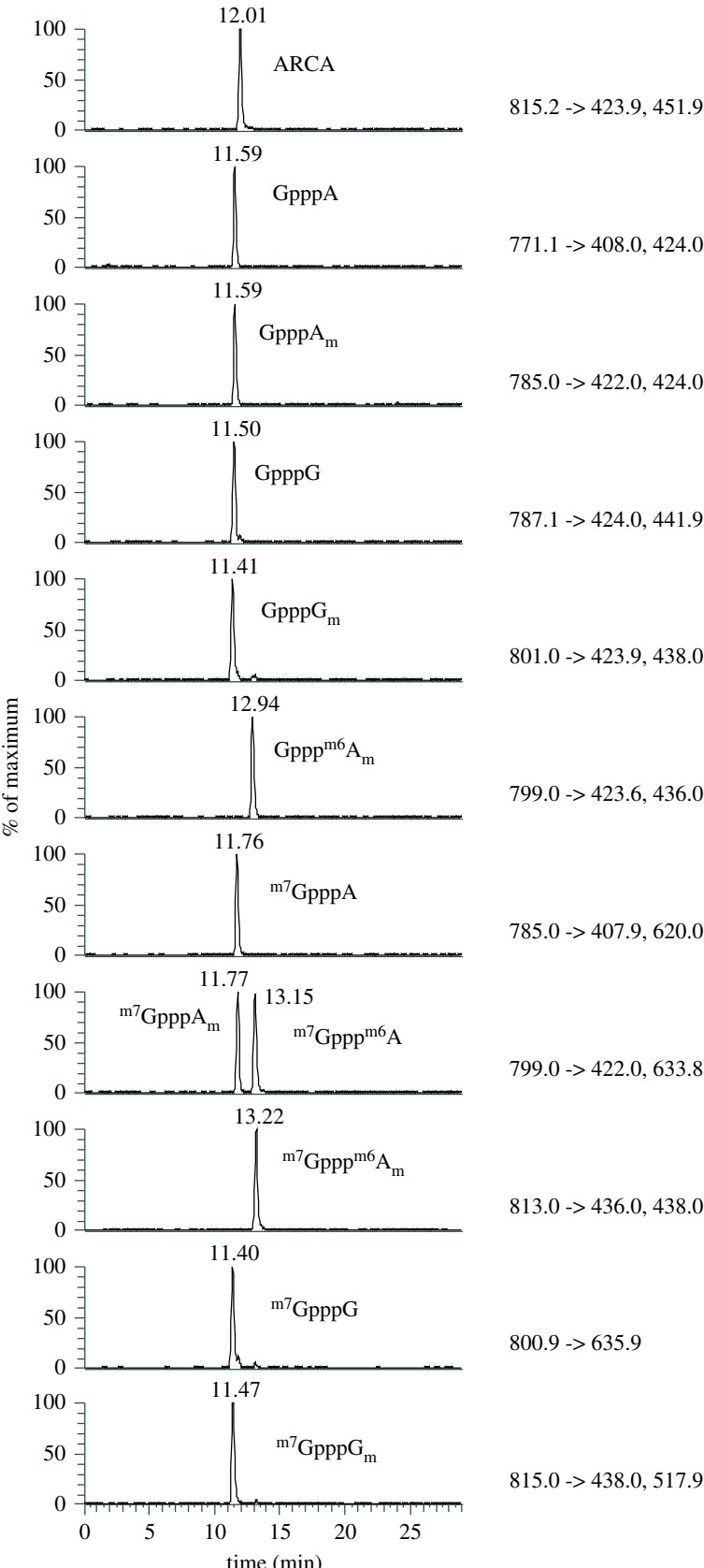

**Figure 3.** Elution profiles of 11 cap nucleotides on a PGC column. Chromatogram showing the differential separation of 11 cap nucleotides on a PGC column. MRM values are indicated to the right of each chromatogram. In the lane containing $^{m7}$GpppA$_m$ and $^{m7}$Gppp$^{m6}$A the $^{m7}$GpppA$_m$ elutes earliest.

From the cap structures which we investigated (those with first nucleotide G or A), the major cap structures, $^{m7}$GpppG$_m$, $^{m7}$GpppA$_m$ and $^{m7}$Gppp$^{m6}$A$_m$, were detected in 250 ng mRNA (oligo dT-purified) from tissue sources. Minor, undermethylated cap structures were detected in 15 µg mRNA. As reported by earlier studies, caps methylated on

the N-7 position of the terminal guanosine and O-2 position of the first transcribed nucleotide ribose were the most common across a range of tissue sources/cell types [1,15,18,19]. However, in 2–5% oligo dT-purified transcripts, a variety of incompletely methylated cap dinucleotide structures in which one or both methyl groups were absent were

royalsocietypublishing.org/journal/rsob  Open Biol. **10**: 190306

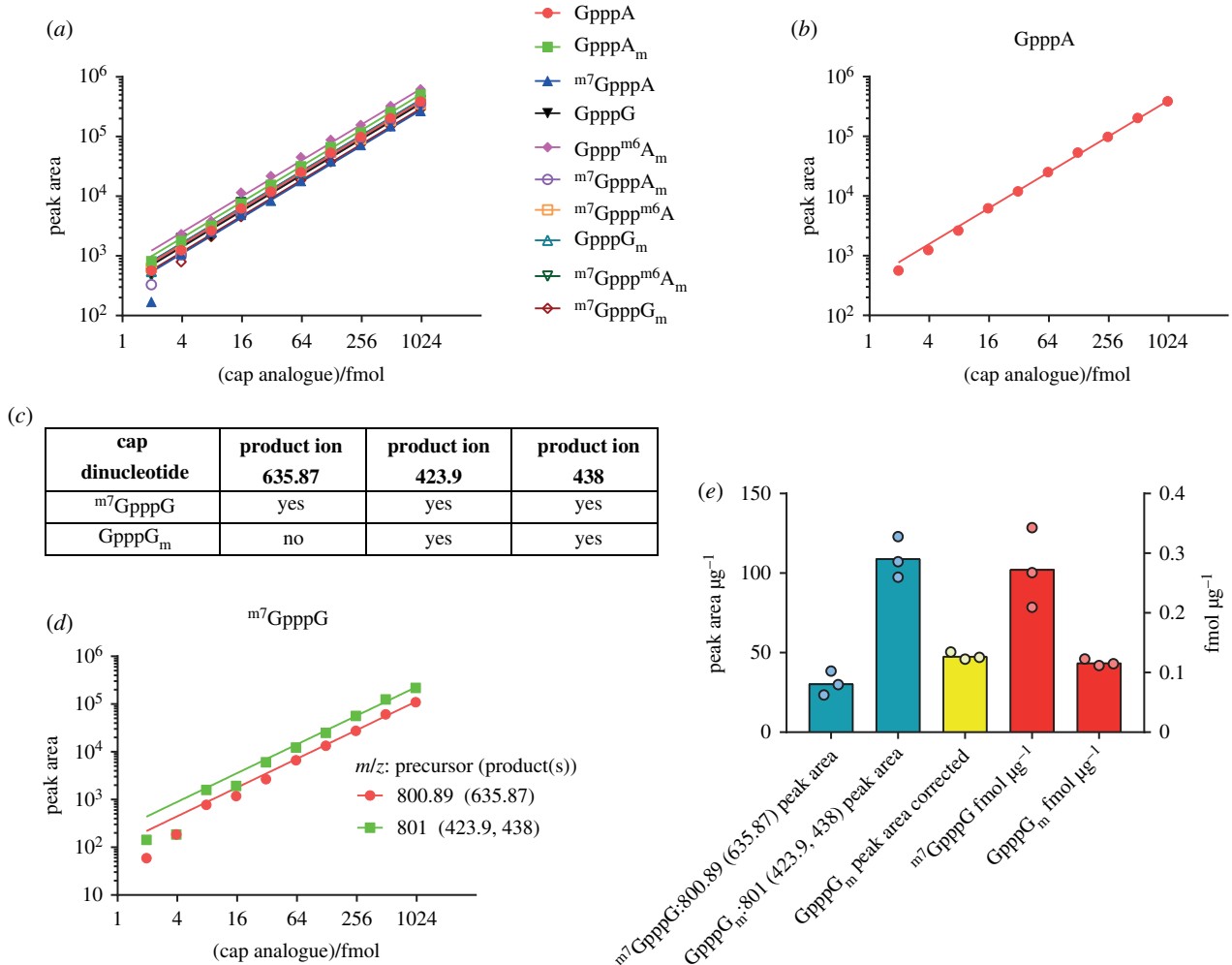

**Figure 4.** Detection of cap dinucleotides in serial dilution. (*a*) Peak area measurements from a serial dilution of synthetic cap dinucleotides. Slopes from linear regression of these values were calculated to allow conversion between peak area and fmol (table 3). (*b*) Peak area measurements for GpppA from a serial dilution of synthetic cap dinucleotides, provided as an example. (*c*) Table demonstrating overlap in product ions originating from $^{m7}$GpppG and GpppG$_m$. (*d*) Detection of $^{m7}$GpppG across a dilution series. $^{m7}$GpppG is detected with its unique *m/z* 800.9 → 635.9 precursor → product ion transition, but it also contributes to the *m/z* 801.0 → 423.9 and 438.0 precursor → product ion transitions shared with the isobaric dinucleotide GpppG$_m$. The *m/z* 801 → 423.9 and 438 transition ion current signals from $^{m7}$GpppG can be back-calculated from the *m/z* 800.9 → 635.9 precursor → product ion signal and subtracted from the total *m/z* 800.9 → 635.9 precursor → product ion signal to allow quantification of GpppG$_m$. (*e*) Compensation for shared ions between $^{m7}$GpppG and GpppG$_m$. $^{m7}$GpppG and GpppG$_m$ raw peak areas are shown and the GpppG$_m$ peak area after correcting for $^{m7}$GpppG forming product ions at *m/z* 423.9 and 438. Femtomoles of cap per µg input mRNA are calculated using the linear fit between fmol and peak area (table 3). Each point represents a biological replicate.

detected. Demethylase activities which target the methyl groups on the N-7 position of the cap guanosine or the O-2 position of the first transcribed nucleotide ribose have not been isolated from mammalian cells and therefore the undermethylated structures are unlikely to be breakdown products. This implies that either a certain amount of mRNA is processed into polyadenylated transcripts in the absence of a completed cap structure and/or the cap is completely removed, recapped and partially methylated [45]. Undermethylated caps were uncommon, typically making up less than 5% of cellular mRNA caps, indicating either that they are efficiently converted to their methylated forms and/or that they are effectively removed by decapping enzymes including DXO and NUDT16 [46,47].

The sequence of cap methylations is of mechanistic interest when considering regulation of mRNA cap formation. The cap0, cap1, cap2 notation, abbreviating $^{m7}$GpppN, $^{m7}$GpppN$_m$, $^{m7}$GpppN$_m$N$_m$, respectively, can imply that guanosine N-7 methylation occurs prior to first nucleotide ribose O-2 methylation; however, to our knowledge, data have not been presented to suggest an obligatory order to cap methylations

in mammalian cells. We detected both GpppN$_m$ and $^{m7}$GpppN cap structures in a variety of tissues, suggesting that RNMT and CMTR1 can independently methylate the RNA cap. This is consistent with *in vitro* assays investigating the activity of these enzymes on GpppG caps and $^{m7}$GpppG caps [48,49]. Notably, CMTR1 binds directly to the RNAPII C-terminal domain whereas RNMT interacts indirectly with RNAPII predominantly via interactions with RNA and RNAPII-associated proteins, which does not enforce or imply an obligate order of action [50,51].

CAPAM/PCIF1, the enzyme catalysing first transcribed nucleotide $^{m6}$A methylation, has recently been identified and characterized [9–12]. $^{m6}$A methylation is the only cap methylation that has been demonstrated to be reversible, leading to an interest in whether it may coordinate signalling events with translation or transcript stability [25]. Consistent with recent estimates of the abundance of the $^{m7}$Gppp$^{m6}$A$_m$ cap, we found it to be more abundant than $^{m7}$GpppA$_m$ in all tissues [9–12]. However, the ratio of $^{m7}$Gppp$^{m6}$A$_m$ to $^{m7}$GpppA$_m$ varied between tissues with higher $^{m7}$Gppp$^{m6}$A$_m$ in heart and brain and lower $^{m7}$Gppp$^{m6}$A$_m$ in liver and CD8 T cells.

**Table 3.** Linear regression cap standards.

| cap | GpppA | GpppA$_m$ | $^{m7}$GpppA | GpppG | Gppp$^{m6}$A$_m$ | $^{m7}$GpppA$_m$ | $^{m7}$Gppp$^{m6}$A | GpppG$_m$ | $^{m7}$Gppp$^{m6}$A$_m$ | $^{m7}$GpppG$_m$ |
|---|---|---|---|---|---|---|---|---|---|---|
| Y Intercept | 0 | 0 | 0 | 0 | 0 | 0 | 0 | 0 | 0 | 0 |
| slope | 393.1 | 503.4 | 275.7 | 356.3 | 626.1 | 365.7 | 358.5 | 410.5 | 412.2 | 292.6 |
| 95% CI slope | 386.9 to 399.3 | 496.5 to 510.4 | 266.2 to 285.2 | 349.8 to 362.7 | 614.2 to 637.9 | 356.7 to 374.8 | 349.5 to 367.4 | 403.5 to 417.4 | 405.1 to 419.3 | 289 to 296.3 |
| $R^2$ | 0.9994 | 0.9995 | 0.9969 | 0.9992 | 0.9991 | 0.9984 | 0.9984 | 0.9993 | 0.9992 | 0.9996 |

| | $^{m7}$GpppG | |
|---|---|---|
| cap/MRM values: | 800.9 (635.9) | 801.0 (423.9, 438.0) |
| Y Intercept | 0 | 0 |
| slope | 112.0 | 224.2 |
| 95% CI (slope) | 108.5 to 115.5 | 214.4 to 233.9 |
| $R^2$ | 0.9975 | 0.9953 |

This indicates that adenosine N-6 methylation by CAPAM or demethylation by FTO is differentially regulated across these tissues. CAP-MAP was able to detect a substantial decrease in $^{m7}$Gppp$^{m6}$A$_m$ levels and an increase in $^{m7}$GpppA$_m$ in HeLa cells following short-term knock down of CAPAM, illustrating its potential to investigate the effects of CAPAM and FTO regulation on cellular cap structures.

Recently, an alternative LC–MS method for detecting mRNA caps was reported by Wang et al. [52]. There are a number of differences between our CAP-MAP approach and that of Wang et al. to be considered when selecting a method of cap analysis. Wang et al. extended their analysis to include a variety of rare non-canonical metabolite mRNA caps, including NAD-caps, which have only recently been described in mammalian cells, and FAD-caps, UDP-Gluc-caps and UDP-GlucNAc-caps, which were not previously observed in mammals [53]. Thus, Wang et al.'s cap analysis methodology will be critical for understanding the physiological functions of metabolite caps in mammalian cells [54]. Since the interest of our laboratory centres on cap methyltransferases, our analysis focused on dinucleotide caps. The main advantage of our approach is that we quantify caps directly from oligo dT-purified RNA digested with nuclease P1, whereas Wang et al. used an additional HPLC purification step to separate cap structures from nucleotide monophosphates. This additional step increases the protocol length and may result in some sample loss; this was carefully controlled for by Wang et al. by the use of heavy isotope-labelled standards for each cap. During HPLC purification of caps, since Wang et al. used a hydrophobic C18 column, the ion-pairing agent DBAA was used to enable hydrophilic cap structures to interact with the column. We wanted to avoid using an ion-pairing agent because it is hard to remove and any remaining in the sample following preparation causes ion suppression, thus reducing MS sensitivity. In addition, because ion-pairing agents are difficult to completely remove from the analytical systems it is an unpopular choice for our shared equipment. Therefore, our CAP-MAP method may be appropriate for those who are using shared equipment and are solely interested in detecting $^{(m7)}$GpppX$_{(m)}$ structures. Those interested in metabolite caps and who have dedicated instruments may find the method described by Wang et al. to be most useful.

There were some differences in the cap dinucleotide structures detected by our methods. We detected a range of undermethylated caps, as discussed above, which were not detected by Wang et al. However, Wang et al. detected substantial amounts of $^{m7}$Gppp$^{m6}$A, similar in concentration to $^{m7}$GpppA$_m$, in mouse liver and the human CCRF-SB cell line mRNA. CAPAM was determined to have a 7.6-fold preference for ribose O-2 methylated caps over ribose unmethylated caps, a selectivity which would permit the formation of $^{m7}$Gppp$^{m6}$A in vivo [10]. In previous analyses of HEK293T and MEL624 cells, $^{m7}$Gppp$^{m6}$A was not detected [9–12]. In CAP-MAP, we readily detected synthetic $^{m7}$Gppp$^{m6}$A, but did not detect this cap dinucleotide in liver, other mouse tissues or HeLa cells. This could reflect either biological differences in the mRNA cap structures present in our samples or differences in our separation and detection of $^{m7}$Gppp$^{m6}$A. In addition to the methodological differences stated above, Wang et al. detected cap dinucleotides in the positive ion mode, whereas we used the negative ion mode; thus the product ions used to confirm the identities of cap dinucleotides in our experiments are different: we used product ions at $m/z$ 422.0 and 633.8 to detect $^{m7}$Gppp$^{m6}$A, whereas Wang et al. used a product ion at $m/z$

royalsocietypublishing.org/journal/rsob    Open Biol. **10**: 190306

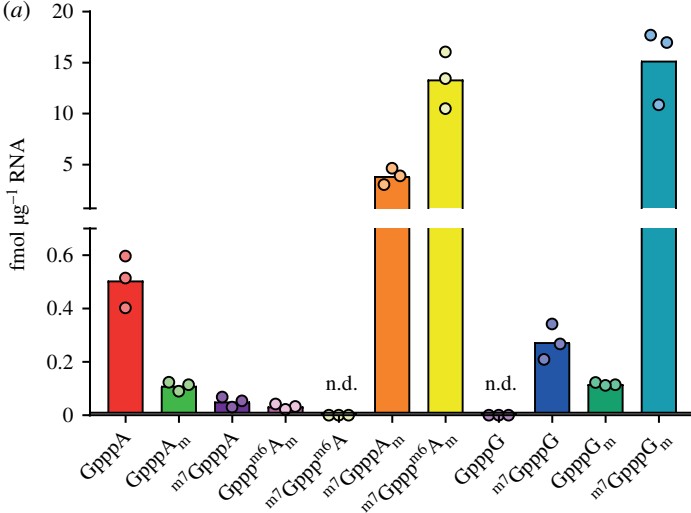

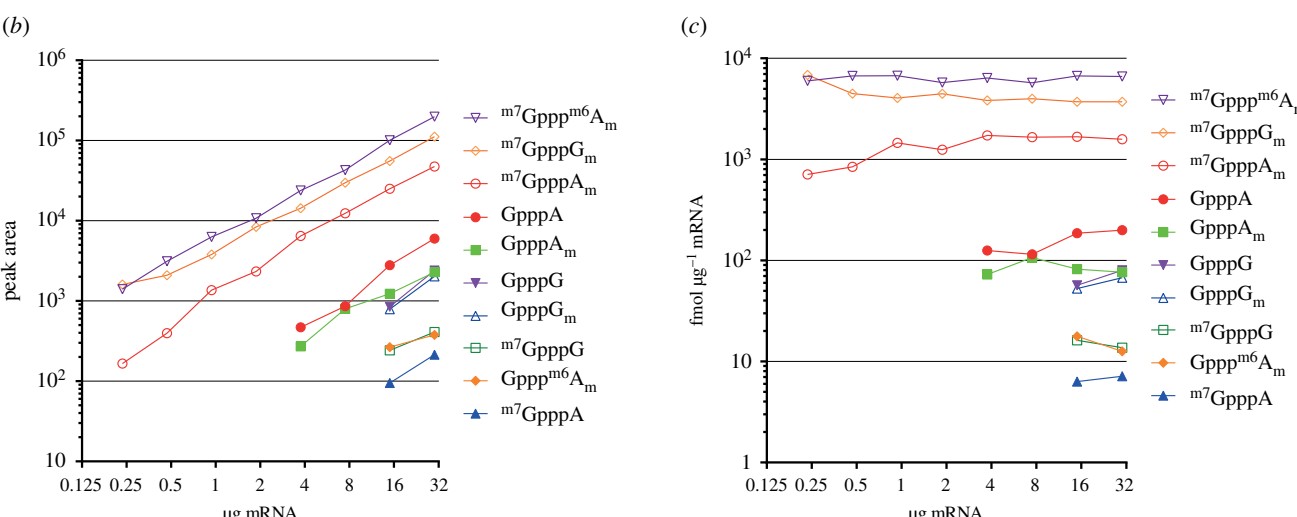

**Figure 5.** mRNA cap dinucleotides detected in mouse liver. (*a*) Abundance of mRNA cap dinucleotides isolated from mouse liver. Each point indicates a biological replicate. n.d. indicates that a cap dinucleotide was not detected. (*b*) Peak area measurements of different mRNA cap dinucleotides in a serial dilution of mouse liver mRNA. (*c*) Calculated relative abundance (in fmol $\mu g^{-1}$ mRNA) of different cap dinucleotides in a serial dilution of mouse mRNA.

150. These differences may modify the sensitivity of detection of cap dinucleotides in the sample, affecting thresholds of detection.

To conclude, as it is becoming more apparent that mRNA cap methylation is a regulated process, a simple and sensitive method for quantifying the mRNA cap structures present in cells is required. Here, we present CAP-MAP, a rapid and direct method for mRNA cap analysis by LC–MS. In CAP-MAP, major cap dinucleotide variants are detected and quantified with low-input mRNA (500 ng), which can rapidly be recovered from cultured cells or tissue samples, and minor cap dinucleotides can be detected using a larger quantity of mRNA.

# 4. Material and methods

## 4.1. Cap dinucleotide standards

Synthetic cap dinucleotide standards were either bought from New England Biolabs (NEB) or synthesized. A list of the cap dinucleotides and their sources is available in table 1. Syntheses of dinucleotide cap analogues $GpppG_m$, $GpppA_m$, $Gppp^{m6}A_m$, $^{m7}GpppG_m$, $^{m7}GpppA_m$, $^{m7}Gppp^{m6}A$,

$^{m7}Gppp^{m6}A_m$ were performed from GDP or $m^7GDP$ and imidazolide derivatives of 2′-O-methylguanosine 5′-O-monophosphate ($m^{2'-O}GMP$), 2′-O-methyladenosine 5′-O-monophosphate ($m^{2'-O}AMP$) and 6,2′-O-dimethyladenosine 5′-O-monophosphate ($m_2^{6,2'-O}AMP$), respectively, according to the methods reported previously [55,56]. The mononuclotide substrates ($m^{2'-O}GMP$, $m^{2'-O}AMP$, $m^6AMP$ and $m_2^{6,2'-O}AMP$) were prepared from the appropriate nuclosides ($m^{2'-O}Guo$, $m^{2'-O}Ado$, $m^6Ado$ and $m_2^{6,2'-O}Ado$) by 5′-O-phosphorylation according to the procedures described earlier [55,57]. The intermediate 6-methyladenosine ($m^6Ado$) was prepared from 6-chloropurine riboside (Aldrich) according to Johnson *et al*. [58]. 2′-O-Methylated intermediates $m^{2'-O}Guo$, $m^{2'-O}Ado$ and $m_2^{6,2'-O}Ado$ were synthesized using methods described by Robins *et al*. [59].

## 4.2. Cap dinucleotide detection and relative quantification by LC–MS

Eleven different cap dinucleotide standards were used to optimize the LC–MS method for cap detection and quantification. Cap nucleotide levels were measured using a TSQ Quantiva mass spectrometer interfaced with an Ultimate 3000 Liquid Chromatography system (ThermoScientific),

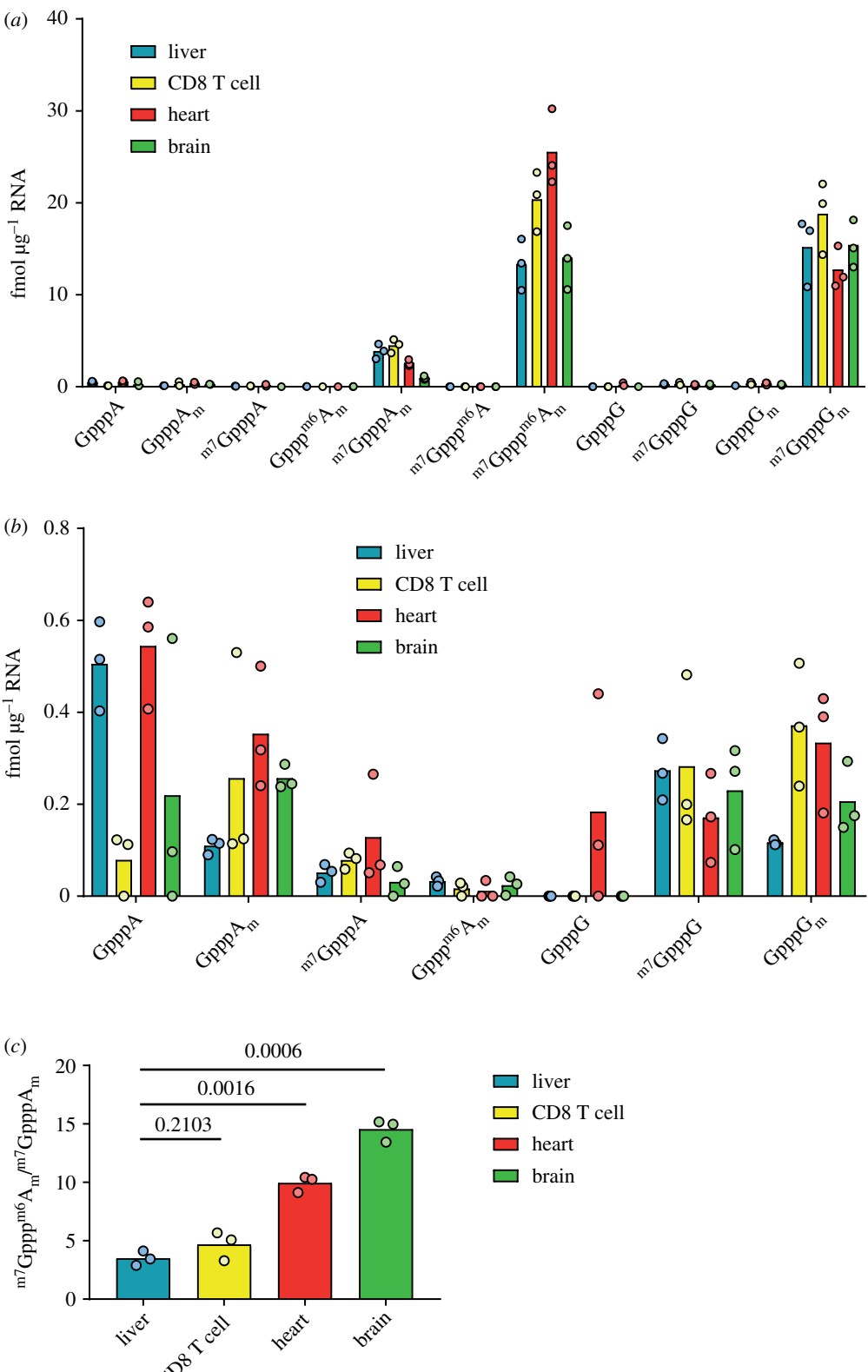

**Figure 6.** mRNA cap dinucleotides detected in mouse organs. (a) Abundance of mRNA cap dinucleotides isolated from mouse liver, activated CD8 T cells, heart and brain. Each point indicates a biological replicate. (b) Data from (a) presented to reveal the abundance of rarer mRNA cap dinucleotides (abundance less than 1 fmol μg$^{-1}$ mRNA). (c) The ratio of $^{m7}$Gppp$^{m6}$A$_m$ to $^{m7}$GpppA$_m$ in mRNA from different sources.

equipped with a PGC column (HyperCarb 30 × 1 mm ID 3 μm; part no. C-35003–031030; Thermo-Scientific). Mobile phase buffer A consisted of 0.3% (v/v) formic acid adjusted to pH 9.15 with ammonia prior to a 1/10 dilution. Mobile phase buffer B was 80% (v/v) acetonitrile. The column was maintained at a controlled temperature of 45°C and was equilibrated with 18% buffer B for 9 min at a constant flow rate of 0.04 ml min$^{-1}$. Aliquots of 14 μl of each sample were loaded

onto the column and compounds were eluted with a linear gradient of 18–21% buffer B over 2 min, 21–40% buffer B over 2 min, 40–60% buffer B over 10 min. The column was washed for 4 min in 100% buffer B before equilibration in 18% buffer B for 9 min. Eluents were sprayed into the TSQ Quantiva using an Ion Max NG ion source with the ion transfer tube temperature at 350°C and vaporizer temperature at 60°C. The TSQ Quantiva was run in negative mode with a

**Table 4.** Summary of cap yields. Where a cap dinucleotide was not detected in one or more replicates this is indicated with n.d.

| cap structure | mean and (s.d.) in fmol µg$^{-1}$ mRNA for each cap structure | | | | | |
| | liver | CD8 T cell | heart | brain | HeLa + si-scrambled | HeLa + si-CAPAM |
| --- | --- | --- | --- | --- | --- | --- |
| $^{m7}$GpppG$_m$ | 15.18 (3.76) | 18.78 (3.97) | 12.74 (2.27) | 15.41 (2.57) | 194.43 (35.09) | 175.60 (19.03) |
| $^{m7}$Gppp$^{m6}$A$_m$ | 13.33 (2.79) | 20.35 (3.26) | 25.52 (4.17) | 14.02 (3.47) | 159.61 (29.11) | 94.36 (12.56) |
| $^{m7}$GpppA$_m$ | 3.86 (0.80) | 4.47 (0.73) | 2.57 (0.34) | 0.96 (0.19) | 17.13 (1.69) | 76.11 (7.44) |
| GpppA$_m$ | 0.11 (0.02) | 0.26 (0.24) | 0.35 (0.13) | 0.26 (0.03) | n.d. | n.d. |
| GpppG$_m$ | 0.12 (0.01) | 0.37 (0.13) | 0.33 (0.13) | 0.21 (0.08) | n.d. | n.d. |
| $^{m7}$GpppG | 0.27 (0.07) | 0.28 (0.17) | 0.17 (0.10) | 0.23 (0.11) | n.d. | n.d. |
| $^{m7}$GpppA | 0.05 (0.02) | 0.08 (0.02) | 0.13 (0.12) | n.d. | n.d. | n.d. |
| GpppA | 0.51 (0.10) | n.d. | 0.54 (0.12) | n.d. | n.d. | n.d. |
| Gppp$^{m6}$A$_m$ | 0.03 (0.01) | n.d. | n.d. | n.d. | n.d. | n.d. |
| GpppG | n.d. | n.d. | n.d. | n.d. | n.d. | n.d. |
| $^{m7}$Gppp$^{m6}$A | n.d. | n.d. | n.d. | n.d. | n.d. | n.d. |

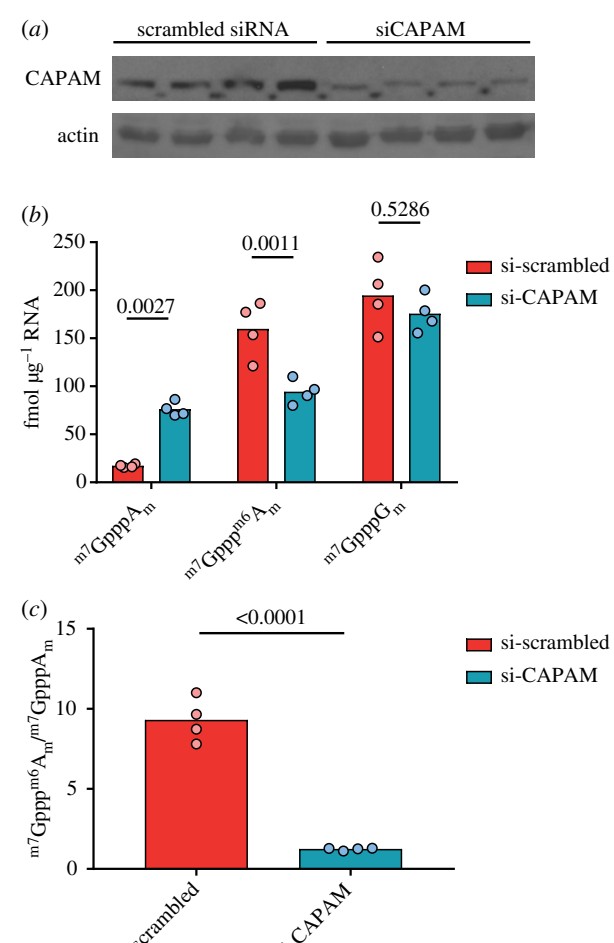

**Figure 7.** Impact of CAPAM knockdown on cap dinucleotide abundance. HeLa cells were transfected with scrambled siRNA or CAPAM siRNA for 3 days. (a) Western blot of CAPAM and actin (from re-probing) in siRNA-treated HeLa cells. (b) Abundance of mRNA cap dinucleotides in siRNA-treated HeLa cells. Samples were compared by an ANOVA, with Sidak's post-test-adjusted p-values are shown. (c) The ratio of $^{m7}$Gppp$^{m6}$A$_m$ to $^{m7}$GpppA$_m$ in mRNA from siRNA-treated HeLa cells. Samples were compared by a Student's t-test.

spray voltage of 3500, Sheath gas 40 and Aux gas 10 and sweep gas 2. Levels of the 11 cap nucleotides were measured using MRM mode with optimized collision energies and radio-frequencies previously determined by infusing pure compounds (table 1). A chemically similar cap standard (ARCA ($^{m7}$G$_{O-3\ m}$pppG)) (NEB) was used as an internal standard (table 1). A standard curve was prepared using 3–1000 fmol of all 11 cap nucleotides with 250 fmol internal ARCA standard spiked into each sample. The standard curve was run prior to running the experimental samples using the same conditions, and was used to calculate the relative amount of the 11 cap nucleotides in each sample. One blank was run between each sample to eliminate carry-over. In one experiment, AMP was detected using $m/z$ 346.0 → 134.2, 151.3, 211.1; these transitions were determined using AMP from Sigma. Representative chromatograms from each experiment are shown in electronic supplementary material, figures S1–S7.

## 4.3. Preparing RNA from tissues

Mice were either C57Bl/6 J mice or control mice from our colonies that are kept on a C57Bl/6 J background. Mice were housed in the University of Dundee transgenic facility. Livers, brains and hearts were snap-frozen in liquid nitrogen, ground in a pestle and mortar and lysed in TRI Reagent (Sigma). RNA was extracted according to the manufacturer's protocol, including the optional centrifugation before the addition of chloroform to remove insoluble material. In addition, a second phenol–chloroform extraction was performed on the aqueous phase from the initial separation in TRI Reagent (ThermoFisher). RNA was dissolved in Hypersolv water (VWR) and quantified using a nanodrop (ThermoFisher).

## 4.4. Preparing RNA from cell cultures

Cells were cultured in 5% CO$_2$, 37°C. CD8 T cells were purified from mouse lymph nodes using an easySep CD8 kit (Stemcell Technologies) and activated on plate-bound anti-CD3 (clone 145–2C11; Biolegend) and anti-CD28 (clone 37.51; Biolegend) antibodies in RPMI medium (Gibco/Thermofisher) with 20 ng ml$^{-1}$ interleukin 2 (Novartis), 10% fetal calf serum (FCS) (Gibco/Thermofisher), 50 mM 2-mercaptoethanol (Sigma) and PenStrep (Thermofisher) for 5 days. Hela cells

were grown in DMEM (Gibco/Thermofisher) with 10% FCS and 5 U ml$^{-1}$ penicillin-streptomycin (ThermoFisher). PCIF1-directed siRNA smartpool or scrambled siRNA (Dharmacon) was transfected using a Neon transfection system (Thermofisher) using 1005 V × 35 ms × 2. CAPAM knockdown was confirmed by western blotting with anti-PCIF1 (ab205016; Abcam) and anti-actin (ab3280; Abcam) antibodies. RNA was prepared from cells as per the tissue samples, with the exception that cells were directly lysed in Tri-Reagent rather than frozen and ground.

## 4.5. Preparing mRNA

For large-scale mRNA extractions, polyadenylated mRNA was enriched using 100 μl oligo dT agarose beads (NEB) per mg RNA. RNA was bound to the beads in binding buffer (1 M NH$_4$OAc, 2 mM EDTA) for 5 min rotating at room temperature and washed once. RNA was eluted in room temperature water, filtered in SpinX (Corning) tubes to remove any beads, then precipitated in 2.5 M NH$_4$OAc and isopropanol, centrifuged at 14 000 r.p.m. × 30 min and pellets washed in 75% ethanol.

To extract mRNA from HeLa cell samples an mRNA Direct kit (ThermoFisher) was used. Magnetic oligo dT beads (40 μl) were used per sample, but with the binding buffer and incubation times indicated above. mRNA was digested with 9.5 units of Nuclease P1 (Sigma) in 20 mM NH$_4$OAC pH 5.3

for 3 h at 37°C. Then, 250 fmol of ARCA was added, and LC–MS was carried out as described above.

## 4.6. Analysis

Peak areas were determined for each transition for each cap dinucleotide. Where there was more than one transition peak on the chromatogram, the correct peak was selected by its retention time relative to the ARCA internal standard and presence of product ions in the ratio indicated by the synthetic standards. The abundances of the various caps from mRNA were calculated by comparison with the dilution series of cap standards. Conversion factors are listed in table 3. Statistical analysis and plots were produced in Prism 5 (GraphPad Software).

Data accessibility. This article has no additional data.

Competing interests. We declare we have no competing interests.

Funding. V.H.C. is supported by funding from the European Research Council (ERC) under the European Union's Horizon 2020 research and innovation programme (grant agreement no. 769080); Medical Research Council Senior Fellowship (MR/K024213/1), Royal Society Wolfson Research Merit Award (WRM\R1\180008), a Lister Institute Prize Fellowship and Wellcome Trust Centre Award (097945/Z/11/Z). M.A.J.F. is supported by a Wellcome Investigator Award (101842/Z13/Z).

Acknowledgements. We thank members of the Cowling laboratory and FingerPrints Proteomics Facility for useful discussions.

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
