## [Reviewer comments · Open Biology]

Review History

RSOB-19-0306.R0 (Original submission)

Review form: Reviewer 1

Recommendation

Major revision is needed (please make suggestions in comments)

Do you have any ethical concerns with this paper?

No

Comments to the Author

The authors report an efficient method for quantifying different mRNA cap structures, via LC-MS. It allows monitoring variation of these cap structure levels in diverse cell circumstances, including viral infection, cell stress, gene knockdown, human disease etc. The method can be quite useful for the scientific community. I have a few questions that will need to be addressed.

In Figure 1., guanosine base structure within the chemical structure of 2nd transcribed nucleotide is wrong, it should be a C=O double bond without protonation at O6.

Figure 3. Shows all LC-MS channels for all 11 different cap structures by their specific MRM

value and retention time. Considering that each channel shows one single peak for building the normalization curve accurately, why there are 2 peaks in the GpppGm channel? Is this a contamination or I am missing something?

In Figure 3., there are 2 peaks in m7GpppAm/m7Gpppm6A channel, and each peak should be annotated to its corresponding cap structure: which peak is m7GpppAm and which m7Gpppm6A? Also, MRM values should be labeled at each x-axis of the 11 cap structures. If author can add information on m7GpppCm and m7GpppUm (maybe also m7GpppC and m7GpppU), it could be useful for users to follow.

In Figure 5A, in m7GpppAm ~5 fmol/ug RNA vs. m7Gpppm6Am ~12 fmol/ug RNA, CAPAM converts most of m7GpppAm into m7Gpppm6Am. However, in GpppAm 0.1 fmol/ug RNA vs. Gpppm6Am ~0.02 fol/ug RNA (the same trend in Figure 6B), it seems that CAPAM methylation depends on cap m7G methylation? Does RNMT knockdown trigger a dramatic decrease of m6Am level in cell mRNA? Maybe some data and explanation can be added in figure 7 and discussion section. Relationship between RNMT and m6Am will be critical to know.

Review form: Reviewer 2

Recommendation

Accept with minor revision (please list in comments)

Do you have any ethical concerns with this paper?

No

Comments to the Author

Galloway and co-workers have reported on the development of an LC-MS assay amenable at the quantification of the different mRNA caps in the transcriptome. In this assay, mRNA is specifically enriched from total RNA via its poly(A)-tail, digested to nucleoside monophosphates and the trinucleotide cap using nuclease P1 and directly submitted to LC-MS analysis. Addition of an internal standard allowed for direct quantification of eleven distinct cap analogues. Since the methylation status of mRNA caps is thought to be dynamic, the method presented in this work can help understand these dynamics under different cellular conditions and in different tissues.

Comments to the authors:

- The authors mention the recently published work of the Dedon group which is conceptually akin to this work.[1] However, the reader might find it difficult to understand the distinct differences between these methods. Could the authors therefore further elaborate on the conceptual differences between their method and the work of Wang et al.?
- The concentrations of the caps detected in this work are about one order of magnitude lower compared to the concentrations detected by Wang et al. (i.e.: m7Gpppm6Am: 20 fmol/μg RNA vs 425 fmol/μg RNA[1], both in mouse C57BL/6 liver cells). Could the authors comment on these differences?
- We wonder why the authors speculate on the retention time of the nucleoside monophosphates when they should be readily detectable in their LC-MS setup?
- The authors used a PGC column for chromatographic separation of the caps and compared its performance to a HILIC column. Could the authors elaborate on the advantages of the PGC column used in this work? Does it offer better separation than a classical C18 column in a UHPLC setup at the given pH?
- A table showing all the respective cap concentrations in all investigated tissues would serve as a nice summary and overview to the reader. Especially, since in the CD8 T and HeLa cells the concentrations of a few select and not all 11 caps are shown in the figures. (Additional experiment: Cap quantification in human CCRF-SB cells. Can m7Gpppm6Am be found? Wang et al. found m7Gpppm6Am to be amongst the most abundant caps, whereas in the

mouse liver cells (also investigated in this study) its concentration was about 10-fold decreased compared to the most abundant caps.[1])

Comments to the editor:

Overall, we highly recommend publishing this work if the aforementioned points are addressed.

References:

[1] J. Wang, B. L. Alvin Chew, Y. Lai, H. Dong, L. Xu, S. Balamkundu, W. M. Cai, L. Cui, C. F. Liu, X.-Y. Fu, et al., *Nucleic Acids Res.* 2019, 1–45.

Decision letter (RSOB-19-0306.R0)

10-Jan-2020

Dear Professor Cowling,

We are pleased to inform you that your manuscript RSOB-19-0306 entitled "CAP-MAP: Cap Analysis Protocol with Minimal Analyte Processing, a rapid and sensitive approach to analysing mRNA cap structures" has been accepted by the Editor for publication in *Open Biology*. The reviewer(s) have recommended publication, but also suggest some minor revisions to your manuscript. Therefore, we invite you to respond to the reviewer(s)' comments and revise your manuscript.

Please submit the revised version of your manuscript within 7 days. If you do not think you will be able to meet this date please let us know immediately and we can extend this deadline for you.

- 1) A text file of the manuscript (doc, txt, rtf or tex), including the references, tables (including captions) and figure captions. Please remove any tracked changes from the text before submission. PDF files are not an accepted format for the "Main Document".
- 2) A separate electronic file of each figure (tiff, EPS or print-quality PDF preferred). The format should be produced directly from original creation package, or original software format. Please note that PowerPoint files are not accepted.
- 3) Electronic supplementary material: this should be contained in a separate file from the main

text and meet our ESM criteria (see <https://royalsocietypublishing.org/rsob/for-authors#question1>). All supplementary materials accompanying an accepted article will be treated as in their final form. They will be published alongside the paper on the journal website and posted on the online figshare repository. Files on figshare will be made available approximately one week before the accompanying article so that the supplementary material can be attributed a unique DOI.

Online supplementary material will also carry the title and description provided during submission, so please ensure these are accurate and informative. Note that the Royal Society will not edit or typeset supplementary material and it will be hosted as provided. Please ensure that the supplementary material includes the paper details (authors, title, journal name, article DOI). Your article DOI will be 10.1098/rsob.2016[last 4 digits of e.g. 10.1098/rsob.20160049].

4) A media summary: a short non-technical summary (up to 100 words) of the key findings/importance of your manuscript. Please try to write in simple English, avoid jargon, explain the importance of the topic, outline the main implications and describe why this topic is newsworthy.

Images

Data-Sharing

It is a condition of publication that data supporting your paper are made available. Data should be made available either in the electronic supplementary material or through an appropriate repository. Details of how to access data should be included in your paper.

Data accessibility section

Sincerely,

The Open Biology Team
<mailto:openbiology@royalsociety.org>

Reviewer(s)' Comments to Author:

Referee: 1

Comments to the Author(s)

The authors report an efficient method for quantifying different mRNA cap structures, via LC-MS. It allows monitoring variation of these cap structure levels in diverse cell circumstances, including viral infection, cell stress, gene knockdown, human disease etc. The method can be quite useful for the scientific community. I have a few questions that will need to be addressed.

In Figure 1., guanosine base structure within the chemical structure of 2nd transcribed nucleotide is wrong, it should be a C=O double bond without protonation at O6.

Figure 3. Shows all LC-MS channels for all 11 different cap structures by their specific MRM value and retention time. Considering that each channel shows one single peak for building the normalization curve accurately, why there are 2 peaks in the GpppGm channel? Is this a contamination or I am missing something?

In Figure 3., there are 2 peaks in m7GpppAm/m7Gpppm6A channel, and each peak should be annotated to its corresponding cap structure: which peak is m7GpppAm and which m7Gpppm6A? Also, MRM values should be labeled at each x-axis of the 11 cap structures.

If author can add information on m7GpppCm and m7GpppUm (maybe also m7GpppC and m7GpppU), it could be useful for users to follow.

In Figure 5A, in m7GpppAm ~5 fmol/ug RNA vs. m7Gpppm6Am ~12 fmol/ug RNA, CAPAM converts most of m7GpppAm into m7Gpppm6Am. However, in GpppAm 0.1 fmol/ug RNA vs. Gpppm6Am ~0.02 fol/ug RNA (the same trend in Figure 6B), it seems that CAPAM methylation depends on cap m7G methylation? Does RNMT knockdown trigger a dramatic decrease of m6Am level in cell mRNA? Maybe some data and explanation can be added in figure 7 and discussion section. Relationship between RNMT and m6Am will be critical to know.

Referee: 2

Comments to the Author(s)

Galloway and co-workers have reported on the development of an LC-MS assay amenable at the quantification of the different mRNA caps in the transcriptome. In this assay, mRNA is specifically enriched from total RNA via its poly(A)-tail, digested to nucleoside monophosphates and the trinucleotide cap using nuclease P1 and directly submitted to LC-MS analysis. Addition of an internal standard allowed for direct quantification of eleven distinct cap analogues. Since the methylation status of mRNA caps is thought to be dynamic, the method presented in this work can help understand these dynamics under different cellular conditions and in different tissues.

Comments to the authors:

- The authors mention the recently published work of the Dedon group which is conceptually akin to this work.[1] However, the reader might find it difficult to understand the distinct differences between these methods. Could the authors therefore further elaborate on the conceptual differences between their method and the work of Wang et al.?
- The concentrations of the caps detected in this work are about one order of magnitude lower compared to the concentrations detected by Wang et al. (i.e.: m7Gpppm6Am: 20 fmol/ μ g RNA vs 425 fmol/ μ g RNA[1], both in mouse C57BL/6 liver cells). Could the authors comment on these differences?
- We wonder why the authors speculate on the retention time of the nucleoside monophosphates when they should be readily detectable in their LC-MS setup?
- The authors used a PGC column for chromatographic separation of the caps and compared its performance to a HILIC column. Could the authors elaborate on the advantages of the PGC column used in this work? Does it offer better separation than a classical C18 column in a UHPLC setup at the given pH?
- A table showing all the respective cap concentrations in all investigated tissues would serve as a nice summary and overview to the reader. Especially, since in the CD8 T and HeLa cells the concentrations of a few select and not all 11 caps are shown in the figures. (Additional experiment: Cap quantification in human CCRF-SB cells. Can m7Gpppm6Am be found? Wang et al. found m7Gpppm6Am to be amongst the most abundant caps, whereas in the mouse liver cells (also investigated in this study) its concentration was about 10-fold decreased compared to the most abundant caps.[1])

Comments to the editor:

Overall, we highly recommend publishing this work if the aforementioned points are addressed.

References:

[1] J. Wang, B. L. Alvin Chew, Y. Lai, H. Dong, L. Xu, S. Balamkundu, W. M. Cai, L. Cui, C. F. Liu, X.-Y. Fu, et al., *Nucleic Acids Res.* 2019, 1–45.

Author's Response to Decision Letter for (RSOB-19-0306.R0)

See Appendix A.

Decision letter (RSOB-19-0306.R1)

30-Jan-2020

Dear Professor Cowling

We are pleased to inform you that your manuscript entitled "CAP-MAP: Cap Analysis Protocol with Minimal Analyte Processing, a rapid and sensitive approach to analysing mRNA cap structures" has been accepted by the Editor for publication in *Open Biology*.

Thank you for your fine contribution. On behalf of the Editors of *Open Biology*, we look forward to your continued contributions to the journal.

Sincerely,
The Open Biology Team
mailto: openbiology@royalsociety.org

Appendix A

Dear Open Biology Team

Thank you for your favourable response to our manuscript.
We have responded to the reviewers comments below.

Changes to the manuscript are marked in green.

Victoria Cowling

Referee: 1

Comments to the Author(s)

The authors report an efficient method for quantifying different mRNA cap structures, via LC-MS. It allows monitoring variation of these cap structure levels in diverse cell circumstances, including viral infection, cell stress, gene knockdown, human disease etc. The method can be quite useful for the scientific community. I have a few questions that will need to be addressed.

Q1: In Figure 1., guanosine base structure within the chemical structure of 2nd transcribed nucleotide is wrong, it should be a C=O double bond without protonation at O6.

Response: Thank you, we have changed this.

Q2: Figure 3. Shows all LC-MS channels for all 11 different cap structures by their specific MRM value and retention time. Considering that each channel shows one single peak for building the normalization curve accurately, why there are 2 peaks in the GpppGm channel? Is this a contamination or I am missing something?

Response: Figure 3 contained representative data from an early run of the PCG column when we were setting up the method and trying a range of columns and buffers. Now, with optimised use, when we run the standards (including when performing titrations of inputs and standards), we have not observed this split peak. In hindsight, it was a mistake to use this example of GpppGm detection for the figure. We have replaced the figure with data from a recent analysis of the standards which is representative of what we observe using the final methodology detailed in the paper.

Q3: In Figure 3, there are 2 peaks in m7GpppAm/m7Gpppm6A channel, and each peak should be annotated to its corresponding cap structure: which peak is m7GpppAm and which m7Gpppm6A? Also, MRM values should be labeled at each x-axis of the 11 cap structures.

Response: The identity of the peaks for m7GpppAm and m7Gpppm6A were indicated in the figure legend, but we have now moved the labels to the figure for clarity and added the MRM values as requested.

Q4: If author can add information on m7GpppCm and m7GpppUm (maybe also m7GpppC and m7GpppU), it could be useful for users to follow.

Response: Yes we agree that information on m7GpppCm and m7GpppUm would be useful. We have added a sentence explaining that we have not yet been able to synthesise these caps and their synthesis intermediates.

Page 5: "These cap dinucleotides have adenosine or guanosine as the first transcribed nucleotide,

thus our method will only detect these variants and not those with cytidine or uridine which are also present in cells.”

Q5: In Figure 5A, in m7GpppAm ~5 fmol/ug RNA vs. m7Gpppm6Am ~12 fmol/ug RNA, CAPAM converts most of m7GpppAm into m7Gpppm6Am. However, in GpppAm 0.1 fmol/ug RNA vs. Gpppm6Am ~0.02 fol/ug RNA (the same trend in Figure 6B), it seems that CAPAM methylation depends on cap m7G methylation? Does RNMT knockdown trigger a dramatic decrease of m6Am level in cell mRNA? Maybe some data and explanation can be added in figure 7 and discussion section. Relationship between RNMT and m6Am will be critical to know.

Response: We agree that this is an interesting point. In Boulias et al, 2019 Mol Cell, N-7 methylation of the cap guanosine was found to be required for efficient CAPAM binding. Suppression of RNMT in cell lines is technically problematic: transient RNMT knock-down does not result in sufficient repression to observe a definitive phenotype and generating rapid degrons/knock-outs is challenging, probably due to the essentiality of the gene.

Referee: 2

Comments to the Author(s)

Galloway and co-workers have reported on the development of an LC-MS assay amenable at the quantification of the different mRNA caps in the transcriptome. In this assay, mRNA is specifically enriched from total RNA via its poly(A)-tail, digested to nucleoside monophosphates and the trinucleotide cap using nuclease P1 and directly submitted to LC-MS analysis. Addition of an internal standard allowed for direct quantification of eleven distinct cap analogues. Since the methylation status of mRNA caps is thought to be dynamic, the method presented in this work can help understand these dynamics under different cellular conditions and in different tissues.

Comments to the authors:

Q1. The authors mention the recently published work of the Dedon group which is conceptually akin to this work.[1] However, the reader might find it difficult to understand the distinct differences between these methods. Could the authors therefore further elaborate on the conceptual differences between their method and the work of Wang et al.?

Response: The Dedon group method differs from ours in sample preparation, LC-MS protocols and in standard preparation and purification. We have added more details regarding the differences between our methodologies in the discussion pages 11 and 12. We focussed on aspects which we think are most likely to affect the choice of method.

Q2: The concentrations of the caps detected in this work are about one order of magnitude lower compared to the concentrations detected by Wang et al. (i.e.: m7Gpppm6Am: 20 fmol/μg RNA vs 425 fmol/μg RNA[1], both in mouse C57BL/6 liver cells). Could the authors comment on these differences?

Response: We did notice that the concentrations of caps detected was lower in our study compared to Wang et al. We also detected more caps/μg mRNA in HeLa cells than the mouse cells/tissues. One probable explanation of the low cap yield, is the lower purity of the mRNA since any contaminating uncapped ribosomal RNA contributes to the RNA signal, but not the cap signal. Notably, for analysis of HeLa cells we used oligo-dT Dynabeads, whereas for the mouse tissues/cells where we had a larger amount (mgs) of total RNA we used oligo-dT agarose beads which have a larger surface for

non-specific binding. Wang et al used oligo-dT Dynabeads for their experiments and in some samples also directly depleted ribosomal RNAs, which probably explains their higher cap yield. Notably within our experiments the cap signal/ μg mRNA is consistent between replicates so although contaminating ribosomal RNA might affect the cap/ μg yield, it does not impact comparisons between different conditions.

Q3: We wonder why the authors speculate on the retention time of the nucleoside monophosphates when they should be readily detectable in their LC-MS setup?

Response: We have corrected this omission by detecting AMP from the nuclease P1 digest. AMP eluted very early from the PGC column, distinctly from the cap dinucleotides. Chromatograms of AMP and other caps are shown in supplemental figure 1, mentioned on page 7.

Q4: The authors used a PGC column for chromatographic separation of the caps and compared its performance to a HILIC column. Could the authors elaborate on the advantages of the PGC column used in this work? Does it offer better separation than a classical C18 column in a UHPLC setup at the given pH?

Response: The HILIC column did not resolve m7Gpppm6A and m7GpppAm, and these are separated well by the PGC column, page 5. We could have used a classical C18 column, but then would need to use an ion-pairing agent in buffers which affects MS sensitivity and is difficult to remove from the system. Wang et al use a classical C18 column with an ion pairing agent to separate cap structures prior to resolution by LC-MS. Wang et al then used a luna omega C18 column which has a positive charge for their LC-MS, although their cap separation was generally good we have noticed there was only 0.4 minutes between the isobaric caps m7GpppAm and m7Gpppm6A, thus the unique product ions they found in the positive ionisation mode were important for distinguishing these caps in their setup. We wanted to use the negative ion mode since the dinucleotide caps have a greater tendency to form negative ions and because of the greater signal to noise ratio, we did not observe any unique product ions between these two cap structures so relied on chromatographic separation.

Q5: A table showing all the respective cap concentrations in all investigated tissues would serve as a nice summary and overview to the reader. Especially, since in the CD8 T and HeLa cells the concentrations of a few select and not all 11 caps are shown in the figures.

Response: Thank you for the good suggestion. The cap concentrations have been added as table 4, discussed on page 8. For the HeLa experiment, other caps were not shown because only 655ng mRNA was analysed due to limits on the RNA yield from cultured cells. The detection of the less common caps isn't possible in 655ng RNA.

(Additional experiment: Cap quantification in human CCRF-SB cells. Can m7Gpppm6Am be found? Wang et al. found m7Gpppm6Am to be amongst the most abundant caps, whereas in the mouse liver cells (also investigated in this study) its concentration was about 10-fold decreased compared to the most abundant caps.[1])

Response: This point is discussed on page 10 and 11, in response to the first question. We assume you refer to m7Gpppm6A caps, which the Dedon study readily observe. We have expanded on our explanation as to why Wang et al might see these caps m7Gpppm6A whereas we do not. Notably we readily detect synthetic m7Gpppm6A and establish its sensitivity of detection. In the Wang et al study, m7Gpppm6A and m7GpppAm were similar in concentration in mouse liver extracts. Although

we don't investigate CCRF-SB cells in which Wang et al saw the most m7Gpppm6A caps, we did look for it in mouse liver extracts and did not detect it, whereas we readily detected m7GpppAm.

Comments to the editor:

Overall, we highly recommend publishing this work if the aforementioned points are addressed.